# Optimization of DNA Fragmentation Techniques to Maximize Coverage Uniformity of Clinically Relevant Genes Using Whole Genome Sequencing

**DOI:** 10.3390/diagnostics15182294

**Published:** 2025-09-10

**Authors:** Vanessa Process, Madana M.R. Ambavaram, Sameer Vasantgadkar, Sushant Khanal, Martina Werner, Maura A. Berkeley, Zachary T. Herbert, Greg Endress, Ulrich Thomann, Eugenio Daviso

**Affiliations:** 1Covaris LLC., a PerkinElmer Company, Woburn, MA 01801, USA; vprocess@covaris.com (V.P.); mambavaram@covaris.com (M.M.A.); svasantgadkar@covaris.com (S.V.); skhanal@covaris.com (S.K.); mwerner@covaris.com (M.W.); gendress@covaris.com (G.E.); uthomann@covaris.com (U.T.); 2Molecular Biology Core Facilities, Dana-Farber Cancer Institute, Boston, MA 02215, USA; maura_berkeley@mail.dfci.harvard.edu (M.A.B.); zherbert@mail.dfci.harvard.edu (Z.T.H.)

**Keywords:** whole genome sequencing (WGS), coverage uniformity, GC-bias, adaptive focused acoustics (AFA) fragmentation, enzymatic fragmentation, variant detection, chromosomal coverage, library preparation, next-generation sequencing (NGS), PCR-free library preparation

## Abstract

**Background:** Coverage uniformity is pivotal in whole genome sequencing (WGS), as uneven read distributions can obscure clinically relevant variants and compromise downstream analyses. While enzyme-based fragmentation methods for WGS library preparation are widely used, they can introduce sequence-specific biases that disproportionately affect high-GC or low-GC regions. Here, we compare four PCR-free WGS library preparation workflows—one employing mechanical fragmentation and three based on enzymatic fragmentation—to assess their impact on coverage uniformity and variant detection. **Results:** Libraries were generated with Coriell NA12878 and DNA isolated from DNA blood, saliva, and formalin-fixed paraffin-embedded (FFPE) samples. Sequencing was performed on an Illumina NovaSeq 6000, followed by alignment to the human reference genome (GRCh38/hg38) and local realignment. We assessed coverage at both chromosomal and gene levels, including 504 clinically relevant genes detected in the TruSight™ Oncology 500 (TSO500) panel. Additionally, we examined the relationship between GC content and normalized coverage, as well as variant detection across high- and low-GC regions. **Conclusions:** Our findings show that mechanical fragmentation yields a more uniform coverage profile across different sample types and across the GC spectrum. Enzymatic workflows, on the other hand, demonstrated more pronounced coverage imbalances, particularly in high-GC regions, potentially affecting the sensitivity of variant detection. This effect was evident in analyses focusing on the TSO500 gene set, where uniform coverage is critical for accurate identification of disease-associated variants and for minimizing false negatives. Downsampling experiments further revealed that mechanical fragmentation maintained lower Single Nucleotide Polymorphism (SNPs) false-negative and false-positive rates at reduced sequencing depths, thereby highlighting the advantages of consistent coverage for resource-efficient WGS. This study introduces a novel framework for evaluating WGS coverage uniformity, providing guidance for optimizing library preparation protocols in clinical and translational research. By quantifying how fragmentation strategies influence coverage depth and variant calling accuracy, laboratories can refine their sequencing workflows to ensure more reliable detection of clinically actionable variants—especially in high-GC regions often implicated in hereditary disease and oncology.

## 1. Introduction

Whole genome sequencing (WGS) has emerged as a powerful analytical strategy to elucidate genetic variants across the entirety of an individual’s genome. Compared with targeted or exome-based approaches, WGS provides a more comprehensive assessment of both coding and noncoding regions, enabling deeper insights into the genetic basis of disease and fostering more accurate genotype–phenotype correlations [1]. Rapid advancements in sequencing technologies over the past decade have significantly reduced costs and turnaround times, expanding the use of WGS from research laboratories into clinical settings [2]. As a result, WGS now stands at the forefront of precision medicine, guiding clinical decision-making in both inherited (germline) and acquired (somatic) disease contexts.

In germline applications, WGS has demonstrated utility in newborn screening, accelerating the identification of potentially life-threatening conditions before the onset of symptoms [3]. Unlike conventional screening panels limited to a predefined set of disorders, WGS comprehensively interrogates the entire genome for pathogenic variants. In doing so, WGS exhibits reduced sensitivity to GC-content bias relative to WES, enables more uniform coverage, and robustly captures both exonic and noncoding pathogenic variants, as well as SNPs, copy number variants, inversions, and small insertions or deletions [4]. In terms of hereditary disease, WGS offers unparalleled diagnostic power in detecting large deletions and variants in noncoding regions of the genome that might be missed by more targeted methods [5].

Beyond diagnostics, the application of WGS in pharmacogenomics promises to optimize medication regimens by identifying actionable genetic variants associated with drug adsorption, distribution, metabolism, elimination, and toxicity (ADME/Tox). In addition, pharmacodynamics or drug efficacy can be strongly impacted by certain genotypes as well [6]. Several genes carry polymorphisms that have well-established links to adverse drug reactions or therapeutic failure; incorporating WGS data into clinical practice can help reduce trial-and-error prescribing and improve patient outcomes [7].

NGS applications in somatic oncology have similarly benefited from the comprehensive insights afforded by WGS. In the realm of solid tumors and hematologic malignancies, WGS can pinpoint tumor-specific tier-1 mutations such as structural variations and copy number alterations—information that can be leveraged to select targeted therapies and monitor disease progression [8]. Furthermore, WGS aids in minimal residual disease (MRD) monitoring by enabling the detection of rare tumor-specific genetic markers that persist after therapy, thereby allowing for timely intervention upon disease recurrence. In line with this approach, technologies such as PhasED-seq—initially validated for B-cell malignancies—can be extended to solid tumors by integrating up-front WGS of tumor–normal pairs to design personalized, phased variant -enriched panels [9]. The transformative potential of WGS is a single, unified test that supports both germline and somatic analyses for a wide array of clinical indications spanning newborn health, hereditary disorders, pharmacogenomics, and oncology.

Multiple sample prep and sequencing derived performance metrics are crucial to ensure the reliability and interpretability of next-generation sequencing (NGS) results, including sensitivity, specificity, accuracy, and coverage uniformity [10]. Coverage uniformity refers to the consistency of read-depth across the regions of interest (e.g., an exome or entire genome). High and uniform coverage helps reduce false negatives, especially in challenging genomic regions characterized by high-GC content or repetitive sequences and is central to accurate variant detection and downstream analyses [11]. In a clinical setting, insufficient or uneven coverage can lead to missed pathogenic variants and ultimately compromise patient care.

Targeted panels leveraging hybrid-capture NGS workflows have been extensively employed to evaluate exome coverage, where differences in probe design often lead to coverage inconsistencies. These systematic biases hinder the integration of datasets across platforms, complicating the detection of single-nucleotide variants (SNVs) and copy number variations (CNVs) [12]. However, in these studies, the potential influence of DNA fragmentation outcomes during library preparation was not examined.

Advances in genomic technologies have enhanced the detection and interpretation of CNVs and structural variants (SVs) across prenatal, postnatal, and population studies. In prenatal diagnostics, chromosomal microarray analysis (CMA) is widely employed for CNV detection, offering an approximate 6% improvement in diagnostic yield compared with conventional karyotyping [13]. The complementary use of WES or WGS enables identification of additional variants, including single nucleotide variants, small insertions and deletions (indels), and CNVs not detected by CMA [13]. In postnatal and clinical applications, comprehensive CNV and SV characterization is essential for diagnosing genetic disorders, informing prognosis, and guiding therapeutic decisions. Large-scale resources such as Genome Aggregation Database have cataloged over 430,000 SVs from nearly 15,000 human genomes, revealing their significant contribution to protein-disrupting changes and disease susceptibility [14]. In population studies, large-scale analyses using non-invasive prenatal testing have revealed thousands of CNVs across regional cohorts, demonstrating the utility of clinical genomics for estimating population-level variant frequencies [15]. Furthermore, global initiatives such as the 1000 Genomes Project [16] and the Human Pangenome Reference [17] are enhancing SV detection sensitivity and accuracy, thereby providing a more comprehensive understanding of human genomic diversity. Collectively, these findings highlight CNVs and SVs as fundamental components of the human genome, with broad implications for prenatal screening, postnatal clinical diagnosis, and population genomics.

Enzyme-based fragmentation methods, such as endonuclease digestion and tagmentation, are known to introduce sequence-dependent biases that can affect downstream analyses. For instance, tagmentation by transposases like Tn5 may preferentially cleave lower-GC regions, leading to an imbalance in coverage between low-GC and high-GC segments [18]. Similarly, specific endonucleases can display nucleotide-sequence preferences, further contributing to non-uniform representation across the genome [19]. The uniformity of coverage in WGS is a key quality metric for ensuring accurate variant detection, minimizing bias, and improving the reliability of downstream applications. Uneven sequence read coverage raises significant concerns for downstream analysis, with key influencing factors including GC content, coverage uniformity, and read depth [12]. In this study, we assess coverage uniformity of chromosomal regions and genes of clinical relevance prepared with enzymatic fragmentation and mechanical shearing of DNA from Coriell NA12878, blood, saliva, and FFPE.

## 2. Materials and Methods

### 2.1. Sample Collection and Extraction

EDTA whole-blood and saliva samples were collected by Precision for Medicine (Norton, MA, USA). Genomic DNA (hgDNA) from blood was extracted using the KingFisher Duo-Ready DNA Ultra 2.0 Prefilled Plates (Cat. No. A36584, Thermo Fisher Scientific, Waltham, MA, USA) according to the manufacturer’s instructions, with the exception that 10 mM Tris pH 8 was used as the elution buffer instead of the kit’s default buffer. Saliva hgDNA was extracted using the QIAamp DNA Midi Kit (Cat. No. 51183, Qiagen, Germantown, MD, USA) according to the manufacturer’s instructions. FFPE from Lung tissue was acquired from the Cooperative Human Tissue Network (CHTN, Philadelphia, PA, USA). gDNA from this FFPE tissue was extracted using the truXTRAC FFPE Total NA Auto 96 Kit (Cat. No. 520318, Covaris, Woburn, MA, USA).

### 2.2. Cell Lines

Genomic DNA from the B-lymphocyte-derived human cell line NA12878 was obtained from the Coriell Institute for Medical Research (Camden, NJ, USA; https://www.coriell.org). NA12878 is a well-characterized and widely used reference standard in genomic research. As the DNA was sourced from a publicly available and established cell line repository, no additional institutional review board or ethics committee approval was required.

### 2.3. Library Preparation and Sequencing

Libraries were prepared using four different WGS PCR-free library prep kits from various sample types, including genomic DNA from the lymphoblastoid cell line NA12878 (Coriell Institute for Medical Research, Camden, NJ, USA), blood, saliva, and FFPE lung tissue (Precision for Medicine, Frederick, MD, USA). Libraries were prepared using the truCOVER PCR-free Library Prep Kit (Cat. No. 520361, Covaris, Woburn, MA, USA), which utilizes mechanical fragmentation, along with three kits that utilize non-mechanical fragmentation. These three kits included: on-bead tagmentation-based Illumina DNA PCR-Free Prep, Tagmentation (Cat. No. 20041795, Illumina, San Diego, CA, USA), enzymatic-based NEBNext Ultra II FS DNA PCR-free Library Prep Kit (Cat. No. E7430L, New England Biolabs, Ipswich, MA, USA), and enzymatic-based Watchmaker DNA Library Prep Kit with Fragmentation, PCR-free (Cat. No. 7K0013-096, Watchmaker, Boulder, CO, USA).

All the PCR-free library preparation kits were initially optimized using Coriell DNA to obtain a sequencing insert size of ~350 bp, before moving on to the more complex sample types. For the final dataset we followed the fragmentation settings in the published protocols for Covaris, Illumina and NEB. However, the Watchmaker kit required optimization of both time and temperature according to the protocol. Using Coriell DNA, we determined optimal conditions of 37 °C for 4.5 min which were applied to the final dataset. For indexing the Illumina libraires, Index 1 (i7) and Index 2 (i5) adapters from Illumina UD indexes (Cat. No. 20091654, Illumina, San Diego, CA, USA) were used. For all other libraries, the NEBNext Multiplex UDI Oligos for Illumina (Cat. No. E7395S, New England Biolabs, Ipswich, MA, USA) were used.

All libraries, excluding Illumina, utilized the same magnetic bead clean-up after adapter ligation, to minimize the small fragment size bias from Illumina sequencing. The bead clean-up was done using a double clean-up of 0.50× and 0.65× ratios using SPRIselect Bead-Based Reagent (Cat. No. B23318, Beckman Coulter, Brea, CA, USA). Illumina libraries were completed following the manufacturer’s protocol, this includes a double-sided clean-up using the magnetic beads in the kit. All libraries were made using 100 ng input and were completed in three technical replicates for each condition.

All libraries were pooled and sequenced at Dana-Farber Cancer Institute Molecular Biology Core Facilities on a single NovaSeq 6000 S4 flow-cell (Cat. No. 20028312, Illumina, San Diego, CA, USA) with 2 × 150 bp paired-end sequencing.

### 2.4. WGS Analysis and Normalized Coverage

Sequencing reads were processed using CLC Genomics Workbench v.24 (Qiagen, Germantown, MD, USA). FASTQ files were quality-trimmed, aligned to the GRCh38/hg38 reference genome, and locally realigned using CLC’s LightSpeed Germline Variant Module. The resulting read coverage was analyzed using CLC Genomic Workbench’s ‘QC for Targeted Sequencing’ tool and corresponding region files. Chromosomal coverage was analyzed using the hg38 region file, and TSO500 gene coverage was analyzed as described in the next section.

### 2.5. TSO500 DNA Analysis Using CLC Workbench

TSO500 gene coverage was assessed using CLC Genomic Workbench’s target region file ‘tso500_v1.0_hg38_hg38_no_alt_analysis_set’ (v1.0), which includes 527 genes across 7637 regions. The CLC TSO500 gene list has 504 genes, which slightly differs from the 523 genes targeted by Illumina’s TSO500 panel. Specifically, CLC excludes three highly polymorphic HLA genes: *HLA-A*, *HLA-B*, *HLA-C*, due to the challenges in sequencing these regions. Conversely, CLC includes 7 additional genes, *DCC*, *HSPH1*, *PRKG1*, *REEP5*, *SLC7A8*, *STT3A*, and *ZNF2*. This region file is used as part of CLC Interpret (Qiagen) workflow method for TruSight Oncology 500 (Illumina, San Diego, CA, USA) and is tailored to capture probes coverage of the genes of interest targeted by the assay. Data analysis and visualization were performed in RStudio (v2024.09.0) using the tidyverse (v2.0) package. For TSO500 coverage, we filtered out genes on the X chromosome to keep the analysis within the autosomal chromosomes, this resulted in a final list of 504 genes across 7229 regions from the CLC Genomic Workbench TSO500 file (Appendix A). The average coverage and GC content were calculated for each region, and the values were then averaged at the gene level. For the autosomal chromosomes and the TSO500 504 gene datasets, the coverage was normalized to the average coverage within each library kit, sample type, and technical replicate. These normalized coverage values were then averaged across the three technical replicates. Coverage was normalized to correct for the differences due to pooling and sequencing between the sample types and different library kits. For the GC content and normalized coverage scatter plots, a linear regression model was conducted to examine the relationship between the two variables, as well as the R-squared values.

### 2.6. Variant Performance

For variant performance, Coriell NA12878 library reads were downsampled from the FASTQ file sets to approximately 10X (250 million reads) and 20X (500 million reads) using the CLC Genomic Workbench ‘Subsample Sequence List’ tool. The downsampling process balances data depth and performance, with 10X coverage serving as a baseline for variant calling and 20X coverage enhancing confidence, particularly for low-frequency variants and complex genomic regions. The downsampled FASTQ files were then processed using the CLC LightSpeed Germline Variant Module. The resulting variant files were exported in VCF format from CLC and compared to the NIST Genome in a Bottle HG001/hg38 truth set v4.2.1 using hap.py-vceval [20].

## 3. Results

### 3.1. Sequencing Data Quality and Coverage

Before conducting a comprehensive analysis, we first evaluated the sequencing data quality by assessing the percentage of mapped reads to the hg38 reference genome. For Coriell, blood, and FFPE samples, all library preparation workflows achieved a mapping percentage above 99.2%, except for the Illumina workflow with FFPE samples, which had a slightly lower mapping rate of 98.6% (Table 1). In contrast, saliva samples exhibited greater variability, with mapping percentages ranging from 65.6% to 71.0%. This reduction in mapping efficiency is attributed to the presence of non-human contaminant DNA, which is co-extracted from saliva samples during library preparation [21].

To ensure comparative data quality, we analyzed DNA fragment insert sizes, which play a crucial role in sequencing efficiency and downstream bioinformatics analyses. Across all library preparation methods, insert sizes ranged between 336 bp (NEB FFPE sample) and 449 bp (Watchmaker Coriell sample) (Appendix A), confirming they were within the recommended range for WGS and consistent across the different workflows.

We next evaluated the sequencing depth and genome coverage, ensuring they are within an acceptable range for analysis (Table 1). FFPE samples consistently exhibited lower coverage, ranging between 14.6X and 21.6X across all library preparation kits. In contrast, Coriell, blood, and saliva samples achieved higher sequencing depths, ranging from 18.3X to 30.6X.

### 3.2. Base Bias in Sequencing Reads

To assess DNA fragmentation bias, we analyzed the base distribution of the raw reads from the FASTQ files. It has been established that endonucleases and tagmentation-based library preparation kits exhibit base composition biases in the first 15 bases of both Read 1 and Read 2, as these regions correspond to the fragment ends generated during DNA cleavage. However, this pattern does not reflect a random fragmentation process, as documented in [19]. The extent to which this preferential cleavage influences the enrichment or depletion of specific DNA regions remains unclear.

To investigate this, we plotted the percentage of each nucleotide (A, T, G, and C) within the first 20 bases of Read 1 and Read 2 across the library kits (Figure 1). Given that the human genome is approximately 30% adenine (A) and thymine (T) and 20% guanine (G) and cytosine (C), deviations from these proportions indicate potential break point bias.

Our analysis revealed that library preparation workflows using mechanical fragmentation closely reflect the expected human genome base composition, with minimal bias limited to the first 2 bases. In contrast, workflows employing tagmentation or enzymatic fragmentation exhibited pronounced base composition biases across all four DNA bases (Figure 1, Coriell DNA samples; Appendix A, base composition for blood, saliva, and FFPE DNA samples).

Our initial assessment confirmed that mechanical fragmentation provides highly consistent and unbiased results, with libraries from four different vendors all showing base composition distributions that align well with the human genome (Appendix A). This stands in sharp contrast to enzymatic preparations, which exhibit a lesser degree of repeatability in the base distribution at the point of DNA cleavage. Given the high consistency among mechanical ultrasonication methods to shear the DNA, we selected a single library preparation with mechanical fragmentation to serve as a singular benchmark for our main uniformity of coverage analysis. This controlled experimental design allowed us to isolate the fragmentation method as the primary variable and explore whether the known base bias of certain enzymatic kits manifests as uneven sequencing coverage across chromosomes and key genomic regions within genes.

### 3.3. GC Bias Analysis

Based on the differences in fragmentation seen in the raw sequencing reads, we then investigated coverage performance and tested if the base composition differences would also be related to the various GC base content regions of the human genome, and further across chromosome and genes.

We first examined the normalized coverage in relation to GC content across the human genome for the four library kits and sample types. The normalized coverage showed differences between both library kit types and sample types for GC content between 20–80% (Figure 2). For Covaris library prep, uniform coverage was observed across all sample types compared to the enzymatic kits. NEB and Watchmaker library kits showed uniform coverage for blood and Coriell samples, while the coverage was more biased in low-GC (>1.2) and high-GC regions (<0.8) for FFPE and saliva samples. The Illumina library prep showed biased coverage in low-GC and high-GC regions, with similar trends for all sample types. Uniform coverage consistency, particularly in high-GC regions, was observed in mechanically sheared libraries from challenging samples with compromised DNA integrity, such as FFPE and saliva. Notably, FFPE and saliva samples exhibited higher coverage variability with all of the enzymatic fragmentation kits.

### 3.4. Chromosomal Coverage Uniformity Analysis

To assess the impact of GC content on chromosomal coverage, we analyzed normalized coverage across the autosomal chromosomes for all sample types. Figure 3A presents saliva samples as a representative dataset, illustrating distinct chromosomal coverage trends across the different chromosomes. Appendix A shows the corresponding chromosomal coverage profiles for blood, Coriell, and FFPE samples. Chromosome-specific coverage variations observed across all the library kits and sample types. Notably, Chromosomes 1, 13, 16, and 21 exhibited distinct coverage differences, which may be attributed to repetitive regions, such as ribosomal DNA (rRNA) clusters present on chromosomes 13, 14, 15, 21, and 22 [18]. In saliva samples, chromosomes 4, 13, 19, and 22 displayed variability between library kits, with chromosome 19 exhibiting the largest coverage range from 0.94 to 0.83, highlighting differences in sequencing performance.

To further examine the relationship between GC content and sequencing coverage, we plotted normalized chromosomal coverage against chromosomal GC content, alongside their respective linear regression models (Figure 3B). For saliva, Covaris library prep demonstrated the most consistent coverage across all the chromosomes, suggesting reduced GC bias compared to the enzymatic workflows. However, FFPE samples showed significant coverage variability, with lower coverage observed for chromosomes with high-GC content across all library kits (Figure 3B), highlighting the challenges posed by sample degradation. We looked at the R-squared (R^2^) values to quantify the strength of this relationship between average %GC of chromosomes and the corresponding normalized coverage (Figure 3C). For Coriell, Covaris library prep showed R^2^ value of 0.021, while the remaining 3 enzymatic library kits showed higher R^2^ values between 0.087–0.220. For remaining 3 sample types, Covaris library prep showed R^2^ values between 0.125–0.182, while the remaining 3 enzymatic library kits showed higher R^2^ values between 0.219–0.320. In summary, the Covaris workflow showed the least R^2^ values for all sample types.

### 3.5. Uniformity Coverage Analysis of the TSO500 Regions

Based on the difference in coverage seen across chromosomes and the correlation to the GC content between the chromosomes, we next assessed coverage uniformity at the gene level to develop a more detailed coverage analysis method. We used our filtered TSO500 gene file from CLC Genomic Workbench as a representative dataset to investigate the relationship between GC content and gene coverage.

Using the comprehensive TSO500 regions, we calculated the average GC content within each gene and performed a linear regression analysis to evaluate its correlation with coverage (Figure 4A). The results indicate that similar to the chromosomes, there are differences between the normalized coverages for the different sample types and library kits across different GC contents. Additionally, we computed the R-squared (R^2^) values to quantify the strength of this relationship between coverage and GC content (Figure 4B). The Covaris workflow showed an R^2^ of 0.00 for both blood and Coriell samples, compared to a range of 0.190–0.770 observed with other enzymatic library kits. In addition, a weak correlation was observed between GC content and coverage with Covaris library prep for FFPE and saliva samples with R^2^ values between 0.115–0.262. In comparison to this, other library prep kits exhibited a stronger correlation with R^2^ values between 0.352–0.814. In summary, Covaris library prep exhibited minimal GC bias, as indicated by near zero R^2^ values for blood and Coriell samples, and lower R^2^ values (Figure 4B) for FFPE and saliva compared to enzymatic workflows. These findings suggest that Covaris fragmentation effectively mitigates GC bias, ensuring more consistent coverage across diverse sample types.

The range of normalized coverage values of the TSO500 genes were then visualized to compare the different library kits and sample types (Figure 5A). As a metric for uniformity of coverage, we performed a full width at half-maximum (FWHM) analysis to quantitate the widths of the curves. Covaris library prep delivers the most uniform coverage across all sample types, exhibiting a lower FWHM compared to other library kits. This effect is particularly notable in saliva samples, where the FWHM range is 1.7–2.7x smaller (Figure 5B), indicating minimal variability. In contrast, Illumina exhibits the most non-uniform coverage across sample types, with higher FWHM for all sample types suggesting greater variability. Among the sample types analyzed, FFPE and saliva samples showed the widest distribution of coverage, indicating higher variability and potential biases in sequencing representation.

### 3.6. Variant Performance in TSO500 Regions

To evaluate variant performance within the TSO500 regions analyzed for coverage uniformity, we used the Genome in a Bottle (GIAB, NIST v4.2.1) reference dataset and filtered for variants within the high confidence call set of NA12878 (HG001). This filtering identified a limited subset of variants, comprising 549 out of 3.25 million single nucleotide polymorphisms (SNPs) and 11 out of 467,702 insertions/deletions (indels). Due to the low number of indels, subsequent analyses focused exclusively on SNPs.

Among the 504 genes targeted by TSO500 from the CLC Genomics Workbench, 238 (47.2%) contained at least one SNP in the NA12878 reference dataset. To assess the impact of sequencing depth on variant detection across different library preparation kits, we down sampled the original FASTQ files to 500 million (20X coverage) and 250 million (10X coverage) reads. This approach allowed for a direct comparison of variant calling performance while examining the effects of reduced sequencing coverage on detection accuracy.

At 10X coverage, the differences in variant calling performance were more pronounced due to the inherent limitations of lower coverage. The Covaris library preparation kit exhibited the lowest false negative (FN) call rate (Figure 6B and Appendix A), followed by NEB and Watchmaker, both of which also demonstrated low FN rates. In contrast, the Illumina library kit yielded three times the number of false negatives and twice the number of false positives compared to Covaris.

At 20X coverage, variant calling performance improved across all library kits. The performance of Covaris, NEB, and Watchmaker became more comparable, with all three exhibiting low FN and false positive (FP) rates (Figure 6A,B). However, the Illumina kit continued to show elevated FP and FN rates within the TSO500 regions, indicating comparatively lower performance even at higher sequencing depths. The NA12878 variant calling performance metrics show differences among the library preparation workflows, with FN, FP, and TP counts summarized for Covaris, Illumina, NEB, and Watchmaker kits in (Appendix A, indicating better accuracy for Covaris, NEB, and Watchmaker compared with Illumina.

### 3.7. Variant Performance Across GC-Content Regions in the Human Genome

To evaluate variant calling performance across different GC-content regions of the human genome, we analyzed non-filtered Coriell data. The Genome in a Bottle (GIAB) consortium has released region-specific files that can be used in conjunction with the high confidence call sets to gather variant performance across different genomic subsets [22]. The stratification regions of various GC-content regions were used from GIAB (v.3.5), covering a range from 15% to 85% GC, to analyze data from Coriell NA12878 (HG001). Variant calling performance was measured using the F1-score for both SNPs and Indels at sequencing coverages of 10X and 20X.

Recall and Precision metrics, as well as False Negatives and False Positives can be found in the Appendix A. For SNP performance at 20X coverage, Covaris and NEB have more even F1-scores across the GC content regions, with Covaris having the more uniform F1-scores for both 10X and 20X coverage (Figure 7A). At 10X, this uniformity holds for Covaris except for GC > 75%, and for NEB starts decreasing in performance for GC > 30%. Watchmaker SNP F1-scores show a similar trend to NEB with 10X coverage, with decreasing performance with GC > 30%. At 20X the SNP performance for Watchmaker is similar to NEB and Covaris. Illumina has a pronounced decreasing performance trend for SNPs with higher F1-scores at low GC-content and lower F1-scores at high GC-content regions. The trend is more pronounced at 10X, for example, for variants in the 80–85% GC range, the F1-score for Illumina is 0.598 at 10X, which improved to 0.854 at 20X coverage (Figure 7A). This pattern correlates with the strong GC bias observed in gene coverage for Coriell DNA using Illumina (Figure 5B).

In contrast, indel calling exhibited greater variability across the GC content regions for all library kits, highlighting the increased challenge of calling Indels accurately (Figure 7B). Covaris, NEB and Watchmaker show similar indel performance across the GC content regions, for both 10X and 20X coverage. Illumina shows a similar trend as with SNPs, where the indel F1-scores are decreased with higher GC content regions. At 20X, with GC regions between 65–85%, Illumina had the lowest F1-scores across all library kits for indels.

Overall, Covaris library prep consistently demonstrated higher SNP and Indel F1 scores at both 10X and 20X coverage across a broader range of GC content. In contrast, other library prep kits showed mixed results.

## 4. Discussion

Achieving uniform coverage across the human genome remains a central challenge in whole genome sequencing (WGS), particularly in regions with high GC content [23,24]. In this study, we used a unique approach to assess the uniformity of coverage across the chromosomes and TSO500 genes and observed a correlation between the uniformity and the GC content of these genes. We evaluated both enzyme-based and mechanically sheared library preparation strategies and observed that the mechanical fragmentation approach yielded more consistent uniformity of coverage across chromosomes and genes, which is dependent on the CG/AT content. Notably, such coverage consistency is critical for reducing the likelihood of missing pathogenic variants in key genomic loci—many of which reside in high-GC segments [24,25]. By mitigating GC-dependent biases, mechanical fragmentation may thus support more reliable detection of both single-nucleotide and structural variants. Consistent uniform coverage for mechanically sheared libraries was observed in challenging samples with compromised DNA integrity, which was not the case with enzymatic fragmentation kits. This suggests that strategies aiming to minimize fragmentation bias could be particularly beneficial in clinical and diagnostic contexts where sample quality may be suboptimal.

We further examined how coverage depth and GC content influences variant calling accuracy by evaluating downsampled datasets in both genome-wide and gene-level analyses for Coriell DNA. Libraries prepared with mechanical shearing showed comparatively lower rates of false negatives and false positives for single-nucleotide polymorphisms (SNPs), even at lower sequencing depths. Mechanically sheared DNA libraries showed improved SNP F1-scores at both sequencing depths across both low and high GC contents, compared to the enzymatic kit which exhibited biases with higher GC content regions. Meanwhile, variant calling for insertions and deletions (indels) was more variable across all sample types and GC ranges, reflecting the inherent complexity in accurately calling these events [25].

## 5. Conclusions

In addition to demonstrating that mechanical DNA shearing can enhance uniform coverage and reduce GC-related biases, this study introduces a novel framework for assessing coverage uniformity at both the chromosome and gene levels. This study demonstrated that mechanical shearing with Covaris library prep yields a more consistent sequencing coverage across various sample types with minimal impact of GC content. By quantifying how fragmentation strategies affect coverage variability, researchers and clinicians can more accurately estimate the sequencing depth required to achieve consistent representation of clinically relevant genes. In this way, our findings not only deepen our understanding of library preparation-induced biases but also provide a practical tool for optimizing WGS protocols to ensure high-confidence variant detection in both research and clinical contexts.

Future investigations could expand upon our findings by examining how different next-generation sequencing platforms influence fragmentation-associated biases For instance, a comparative study using multiple sequencing technologies could reveal whether certain biases are platform-dependent or arise predominantly from library preparation conditions. In addition, experiments evaluating the performance of various FFPE repair and extraction kits before fragmentation would help clarify how sample preprocessing steps interact with fragmentation protocols to affect coverage uniformity and variant call accuracy. Beyond whole genome sequencing, applying similar approaches to whole-exome sequencing and hybrid-capture designs could illuminate how fragmentation strategies might reduce probe-based capture inefficiencies and coverage gaps in targeted workflows. Furthermore, the methodology discussed here may also have implications for the construction of large-insert genomic libraries, where minimizing handling-induced fragmentation and potential sequence bias is critical to ensure uniform genome coverage. Collectively, these avenues of research could yield more robust guidelines for library preparation and sequencing platform selection, ultimately supporting higher-confidence variant detection across a wide spectrum of clinical and research applications.

## Figures and Tables

**Figure 1 diagnostics-15-02294-f001:**
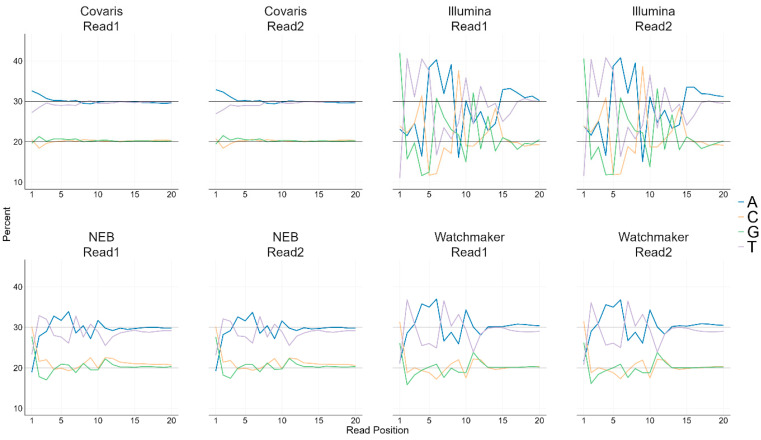
Base Composition of first 20 Sequenced Reads with Coriell DNA. Line plots show the percentage of base composition in the first 20 bases of Read 1 and Read 2 for each library prep kit. Each line represents a different nucleotide base.

**Figure 2 diagnostics-15-02294-f002:**
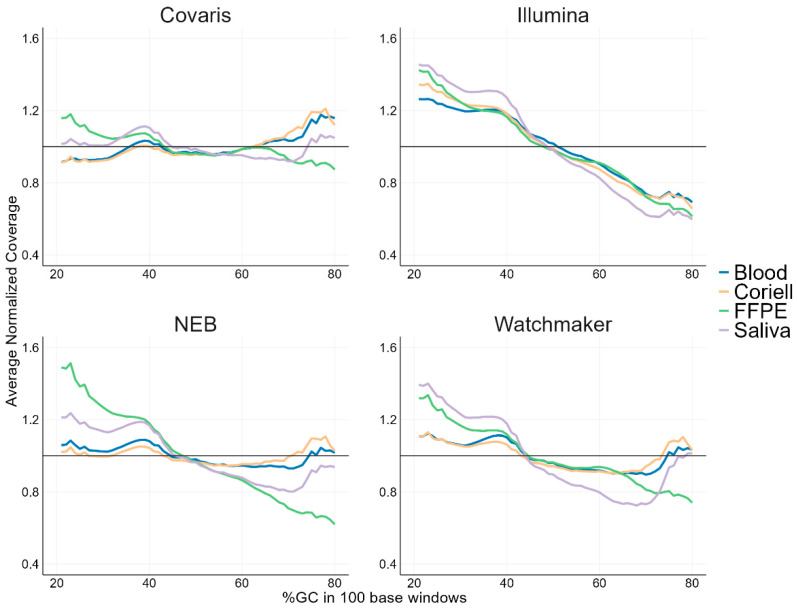
GC-bias line plots showing normalized coverage as a function of GC content across the human genome, calculated in 100 base pair windows. The normalized coverage values shown are from the average of the three technical replicates. Each line represents a different sample type. A normalized coverage value of 1 (shown as a black line) would indicate no GC bias.

**Figure 3 diagnostics-15-02294-f003:**
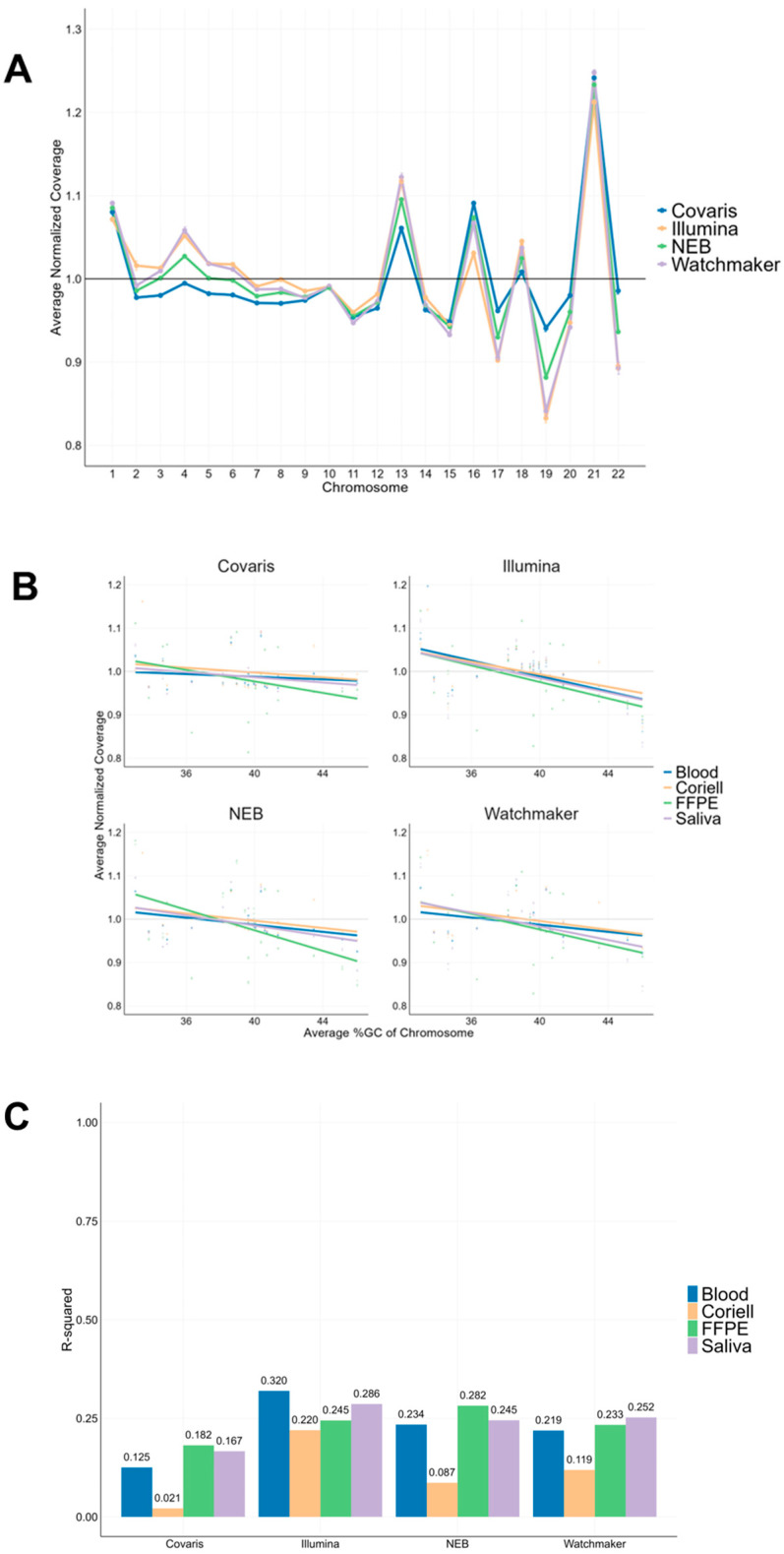
Chromosomal coverage distribution normalized by total coverage. (**A**) Representative line plot of normalized coverage across the autosomal chromosomes for saliva samples, with each line representing a different library kit. X-axis are the 22 autosomal chromosomes. (**B**) Scatterplots showing the relationship between chromosomal %GC content and the normalized coverage for each library kit. Linear regression models were analyzed for the different sample types. A normalized coverage value of 1 (shown as a gray line) would indicate no GC bias. (**C**) Bar plots of R^2^ values for the linear regression models shown in Figure 3B. The plots summarize the relationship between the average %GC of chromosomes and the corresponding normalized coverage. An R^2^ value close to ‘0’ indicates minimal correlation between GC content and coverage.

**Figure 4 diagnostics-15-02294-f004:**
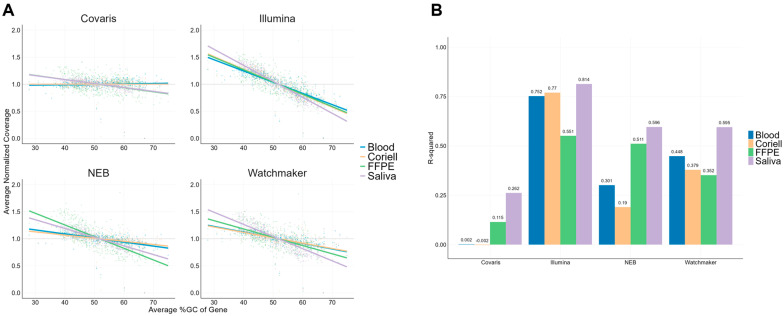
Linear regression analysis of GC content and normalized coverage of the TSO500 genes. (**A**) Linear regression models overlaid to highlight coverage trends, each line represents a different sample type. (**B**) Bar plots showing the R-squared values for the linear regression lines in 4a highlighting relationship of GC content and coverage. An R^2^ value close to ‘0’ indicates minimal correlation between GC content and coverage.

**Figure 5 diagnostics-15-02294-f005:**
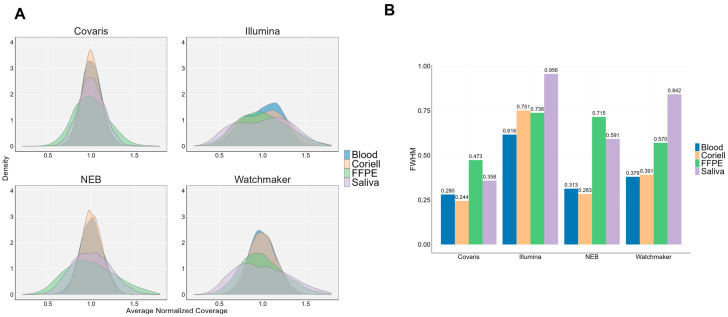
Normalized coverage distributions across the TSO500 genes. (**A**) Histogram plots of average normalized coverage using kernel density estimation, separated by library kit and sample type. (**B**) Full width at half maximum (FWHM) for each density-curve shown in 5A, used as a metric for uniformity of coverage. Lowest FWHM values are observed with Covaris library prep across sample types, exhibiting high coverage uniformity with minimal impact of GC content on coverage.

**Figure 6 diagnostics-15-02294-f006:**
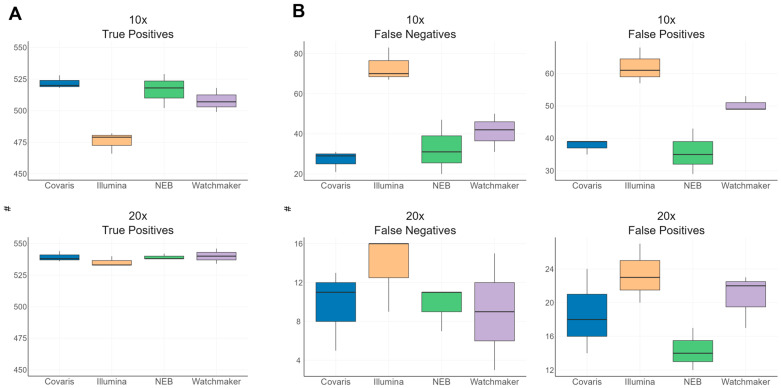
NA12878 Variant Performance within TSO500 regions. Comparison of true and false SNP calls within the TSO500 regions for NA12878 at two different coverage depths across various library kits. The Y-axis represents the count of variant calls. (**A**) Displays true positive SNP calls out of 549, with the top bar plot representing 10X coverage and the bottom bar plot representing 20X coverage. (**B**) Illustrates false negative and false positive SNP calls for the same coverage depths.

**Figure 7 diagnostics-15-02294-f007:**
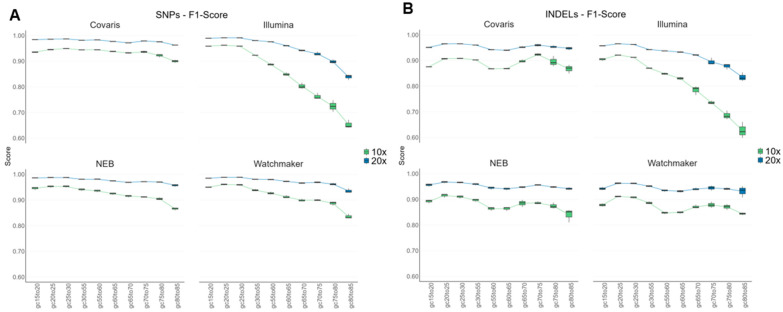
NA12878 Variant Performance across GC content regions, comparison between 10X and 20X coverage across different library kits. The X-axis represents GC content regions ranging from 15 to 85%, with most bins spanning 5% GC intervals, except for the 30–55%GC bin, which covers 20% GC. The individual bins indicate ≥100 bp hg38 regions that fall within the specified GC range, based on the GIAB stratification files. (**A**) F1-score of SNPs. (**B**) F1-score of Indels.

**Table 1 diagnostics-15-02294-t001:** Sequence metrics for the four library preparation kits, all sequenced on the NovaSeq 6000 S4 flow-cell. Each value is an average of the three technical replicates.

Library Prep Kit	Sample	Average Insert Size (bp)	Average of Reads (M)	Mapped Reads (%)	Average Coverage (X)
Covaris	Blood	389	308.8	99.5	25.5
	Coriell	389	339.6	99.5	28.3
	FFPE	348	301.9	99.5	19.7
	Saliva	378	335.9	71.0	24.9
Illumina	Blood	376	372.9	99.1	30.6
	Coriell	348	303.8	99.2	24.9
	FFPE	329	335.0	98.6	21.6
	Saliva	336	402.6	65.9	26.8
NEB	Blood	407	313.7	99.3	25.9
	Coriell	404	361.2	99.5	29.9
	FFPE	336	223.1	99.3	14.9
	Saliva	368	258.4	69.6	18.3
Watchmaker	Blood	442	302.1	99.4	25.6
	Coriell	429	295.6	99.5	25.3
	FFPE	408	280.6	99.4	14.6
	Saliva	380	264.8	66.4	23.8

## Data Availability

The data supporting this study is available from the corresponding author upon reasonable request.

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
