# Peer review of "Optimization of DNA Fragmentation Techniques to Maximize Coverage Uniformity of Clinically Relevant Genes Using Whole Genome Sequencing"

_diagnostics, 2025, doi:10.3390/diagnostics15182294_

Round 1

Reviewer 1 Report

Comments and Suggestions for Authors

Process et al. presented a work demonstrating the superior performance of sequencing data via mechanical fragmentation method on Covaris platform comparing to other three enzymatic-based methods. It is helpful to show the coverage uniformity achieved by Covaris in both chromosomal and gene level, particularly in GC rich regions, when using genome sequencing. There are concerns and points for improvement:

  • Inadequate citations: In the introduction, the authors explained why uniform genome-wide coverage was important and one of the reasons was SV detection. However, the authors only mention about SV/CNV detection in gene fusion. It is also important for the constitutional SV detection such as prenatal and postnatal studies . In addition, although short DNA-insert is commonly used for genome sequencing, uniform read-coverage is also important for large DNA-insert or mate-pair library construction for cases with different indications.
  • Methodology: for Base Bias:
  • In Section 3.2: when checking the Base Bias, only Coriell NA12878 was investigated across different library kits. How about the other three sample types?
  • In Figure S2: base composition bias was compared between different fragmentation methods as well as four mechanical vendors. What is the purpose for that?
  • The main comparison was performed between 1 mechanical fragmentation method (Covaris) and 3 others which included 1 tagmentation-based (illumina) and 2 enzymatic-based (NEB and Watchmaker). The authors should elaborate deeply on why the comparion between 2 mechincal-based and 2 enzymatic-based methods were not performed?
  • In section 3.4:
  • Only saliva sample was selected to assess the impact of GC on chromosomal coverage. What is the reason to choose this sample type?
  • When comparing four library kits and four sample types, coverage variability was observed in saliva and FFPE samples. Any statistically methods can be used to quantify the differences, like R-squared value (which has been shown in Section 3.5, Figure 4B)?
  • In Section 3.6: False negative (FN) and false positive (FP) were used as parameters to assess the performance between different library kits. But there is lacking numbers or percentages to show the evidence.
  • In Section 3.7: description of Figure 7, the bins for GC contents, Line 406 – “except 30 – 50% GC bin”. But in Figure itself, the interval was 30% and next was 55%. Is it a typo or figure error? Same as in Figure S3. A-D.
  • Overall, for the four library kits compared, illumina always showed worst performance in all parameters. Any possible reasons or explanations?

Author Response

  1. Inadequate citations: In the introduction, the authors explained why uniform genome-wide coverage was important and one of the reasons was SV detection. However, the authors only mention SV/CNV detection in gene fusion. It is also important for constitutional SV detection such as prenatal and postnatal studies. In addition, although short DNA-insert is commonly used for genome sequencing, uniform read-coverage is also important for large DNA-insert or mate-pair library construction for cases with different indications.

We agreed with the reviewer's recommendation to broaden the context of our study. To highlight the importance of CNVs and SVs in a wider range of applications, we have added 5 new references [13-17]. These now provide valuable context on their significance in pre- and postnatal studies, as well as in population studies. These additions, we believe, enhance the manuscript by providing a more comprehensive overview of the field and the broader implications of our findings.

  1. In Section 3.2: when checking the Base Bias, only Coriell NA12878 was investigated across different library kits. How about the other three sample types?

We thank the reviewer for this question. We added Figure S2 to show that the base bias fragmentation patterns are conserved across blood, saliva, and FFPE as well.  

  1. In Figure S2: base composition bias was compared between different fragmentation methods as well as four mechanical vendors. What is the purpose for that?

We thank the reviewer for the opportunity to clarify this point. The purpose of comparing four different mechanical fragmentation vendors in Figure S3 was two fold:

  • To Establish a Reliable Control: We first needed to empirically demonstrate that mechanical fragmentation provides a consistent, unbiased, and highly reproducible method for library preparation, regardless of the specific vendor. As shown in Figure S3, all four vendors yielded libraries with base composition distributions that closely mirrored the human genome, confirming their suitability as a "gold standard" control.
  • To Justify Using a Single Benchmark: By proving that different mechanical methods perform almost identically, we justified our decision to use just one of them as our singular benchmark in the main analysis. This created a controlled experimental design, allowing us to confidently attribute any observed coverage deviations in the other libraries directly to the enzymatic fragmentation methods rather than to random variation between controls.

This reasoning has now been explicitly stated in the manuscript (Section 3.2).

  1. The main comparison was performed between 1 mechanical fragmentation method (Covaris) and 3 others which included 1 tagmentation-based (illumina) and 2 enzymatic-based (NEB and Watchmaker). The authors should elaborate deeply on why the comparion between 2 mechincal-based and 2 enzymatic-based methods were not performed?

In connection with the previous point, figure S3 (former S2) shows that mechanical fragmentation methodologies do not show base bias at the point of fragmentation, assuming that this effect would not translate into a base bias in specific genomic regions as shown by both endonucleases and tagmentases. For a focused study we decided to proceed with the bioinformatics of one mechanical based library (truCOVER), with the assumption that all mechanical fragmentation methodologies would show equivalent behavior regarding the base content in DNA. We added a section at the end of section 3.2 to create the connection between the base bias at the point of fragmentation and the uniformity of coverage, which is the key take-away of this study.

  1. In section 3.4: Only saliva sample was selected to assess the impact of GC on chromosomal coverage. What is the reason to choose this sample type?

We thank the reviewer for giving us the opportunity to expand the messaging of the study to every sample we analyzed. We have added data from other sample types in the supporting information (Figure S4).

  1. In section 3.4: When comparing four library kits and four sample types, coverage variability was observed in saliva and FFPE samples. Any statistically methods can be used to quantify the differences, like R-squared value (which has been shown in Section 3.5, Figure 4B)?

We thank the reviewer for pointing out this improvement. We added The R2 value to Figure 3C to improve clarity, modified the figure legend accordingly and added a paragraph in section 3.4.

  1. In Section 3.6: False negative (FN) and false positive (FP) were used as parameters to assess the performance between different library kits. But there is lacking numbers or percentages to show the evidence.

We thank the reviewer for this valuable insight. To quantitatively support our assessment of false negatives (FN) and false positives (FP), we have compiled the relevant data into a new table in the supporting information (Table S2). This table details the performance of each library kit by providing the average number of variants and the standard deviation at sequencing depths of 10x and 20x, thereby offering the numerical evidence requested.

  1. In Section 3.7: description of Figure 7, the bins for GC contents, Line 406 – “except 30 – 50% GC bin”. But in Figure itself, the interval was 30% and next was 55%. Is it a typo or figure error? Same as in Figure S3. A-D.

We thank the reviewer for spotting this inconsistency. We have corrected Figure 7 caption to ensure consistency across figures.

  1. Overall, for the four library kits compared, illumina always showed worst performance in all parameters. Any possible reasons or explanations?

We thank the reviewer for this insightful question. The underperformance of the Illumina kit is most likely attributable to the sequence-dependent bias introduced by its transposase-based fragmentation, a phenomenon that has been previously reported in the literature [23] (previously 18). While our data aligns with this explanation, we have refrained from stating this as a definitive conclusion in the manuscript. The current study was not designed to deconstruct the individual steps of the library preparation process, and therefore, we cannot formally rule out confounding effects from other steps of the workflow.

Reviewer 2 Report

Comments and Suggestions for Authors

The manuscript entitled “Optimization of DNA Fragmentation Techniques to Maximize 2 Coverage Uniformity of Clinically Relevant Genes Using 3 Whole Genome Sequencing”  is a technical study comparing the efficiency of library preparation for high-throughput sequencing. Although it has long been known, and has been shown many times in other publications, the authors have once again confirmed that mechanical fragmentation allows for better results and uniform coverage of the genome.

Unfortunately, most users prefer enzymatic fragmentation, mainly due to the simplicity and speed of the protocols.

Nevertheless, the manuscript is well written and may be useful for scientists involved in high-throughput sequencing. I believe that the article can be published in the Diagnostics journal.

Author Response

  1. The manuscript entitled “Optimization of DNA Fragmentation Techniques to Maximize 2 Coverage Uniformity of Clinically Relevant Genes Using 3 Whole Genome Sequencing” is a technical study comparing the efficiency of library preparation for high-throughput sequencing. Although it has long been known, and has been shown many times in other publications, the authors have once again confirmed that mechanical fragmentation allows for better results and uniform coverage of the genome.

Unfortunately, most users prefer enzymatic fragmentation, mainly due to the simplicity and speed of the protocols.

Nevertheless, the manuscript is well written and may be useful for scientists involved in high-throughput sequencing. I believe that the article can be published in the Diagnostics journal.

We sincerely thank the reviewer for their careful reading and positive assessment of our manuscript. Their encouraging feedback is much appreciated.

Round 2

Reviewer 1 Report

Comments and Suggestions for Authors

The authors have addressed most of my concerns. For the newly added references, the knowledge and reason why this study is also relevent to DNA large-insert library construction can be further enhanced.

Author Response

Reviewer's comment: 

The authors have addressed most of my concerns. For the newly added references, the knowledge and reason why this study is also relevent to DNA large-insert library construction can be further enhanced.

Our reply:

We thank the reviewer for this insightful comment. We agree that our findings have potential implications for large-insert library construction, where maintaining DNA integrity and avoiding representation bias are critical.

The physical forces and fragmentation mechanisms involved in preparing large DNA inserts are indeed different from those in typical short-read library preps, often employing methods like hydrodynamic shearing to minimize excessive fragmentation. While a detailed investigation into how our findings specifically apply to GC/AT bias in those specialized protocols is beyond the scope of this paper, we acknowledge the relevance of this connection.

To address this, we have now added a sentence to the Conclusion section to highlight this point and capture that further studies are needed to quantify and qualify a possible bias that may be due to a different fragmentation mechanism of the DNA: 'Furthermore, the methodology discussed here may also have implications for the construction of large-insert genomic libraries, where minimizing handling-induced fragmentation and potential sequence bias is critical to ensure uniform genome coverage.'